# Feasibility, acceptability and validation of wearable devices for climate change and health research in the low-resource contexts of Burkina Faso and Kenya: Study protocol

Sandra Barteit[1]*, Valentin Boudo[2], Aristide Ouedraogo[2], Pascal Zabré[2], Lucienne Ouremi[2], Ali Sié[2], Stephen Munga[3], David Obor[3], Daniel Kwaro[3], Sophie Huhn[1], Aditi Bunker[1], Rainer Sauerborn[1], Hanns-Christian Gunga[4], Martina A. Maggioni[4,5‡], Till Bärnighausen[1,6,7‡]

1 Heidelberg Institute of Global Health, Heidelberg University Hospital, Heidelberg University, Heidelberg, Germany, 2 Centre de Recherche en Santé, Nouna, Burkina Faso, 3 Kenya Medical Research Institute, Kisumu, Kenya, 4 Institute of Physiology, Center for Space Medicine and extreme Environment Berlin, Charité – Universitätsmedizin Berlin, Corporate Member of Freie Universität Berlin, Humboldt-Universität zu Berlin, and Berlin Institute of Health, Berlin, Germany, 5 Department of Biomedical Sciences for health, Università degli Studi di Milano, Milan, Italy, 6 Department of Global Health and Population, Harvard T.MLP. Chan School of Public Health, Boston, Massachusetts, United States of America, 7 Africa Health Research Institute (AHRI), Durban, KwaZulu-Natal, South Africa

‡ These authors are joint senior authors on this work
* barteit@uni-heidelberg.de

**Funding:** We wish to thank the German Research Foundation (DFG) for supporting this study as part

## Abstract

As the epidemiological transition progresses throughout sub-Saharan Africa, life lived with diseases is an increasingly important part of a population's burden of disease. The burden of disease of climate-sensitive health outcomes is projected to increase considerably within the next decades. Objectively measured, reliable population health data is still limited and is primarily based on perceived illness from recall. Technological advances like non-invasive, consumer-grade wearable devices may play a vital role in alleviating this data gap and in obtaining insights on the disease burden in vulnerable populations, such as heat stress on human cardiovascular response. The overall goal of this study is to investigate whether consumer-grade wearable devices are an acceptable, feasible and valid means to generate data on the individual level in low-resource contexts. Three hundred individuals are recruited from the two study locations in the Nouna health and demographic surveillance system (HDSS), Burkina Faso, and the Siaya HDSS, Kenya. Participants complete a structured questionnaire that comprises question items on acceptability and feasibility under the supervision of trained data collectors. Validity will be evaluated by comparing consumer-grade wearable devices to research-grade devices. Furthermore, we will collect demographic data as well as the data generated by wearable devices. This study will provide insights into the usage of consumer-grade wearable devices to measure individual vital signs in low-resource contexts, such as Burkina Faso and Kenya. Vital signs comprising activity (steps), sleep (duration, quality) and heart rate (hr) are important measures to gain insights on individual behavior and activity patterns in low-resource contexts. These vital signs may be

of the FOR "Climate Change and Health in sub-Saharan Africa". The German Research Foundation (DFG) is supporting this study, but has not been involved in study design, collection, management, analysis or interpretation of data, neither in the writing of this report or in any decision to submit this report for publication (see S2 for informed consent forms). The study has been approved by DFG (FOR 2936 / project: 427397328).

**Competing interests:** The authors have declared that no competing interests exist.

**Abbreviations:** API, Application Programming Interface; CRSN, Centre de Recherche en Santé de Nouna; HDSS, health and demographic surveillance system; hr, heart rate; INDEPTH, International Network for the Demographic Evaluation of Populations and Their Health; KEMRI, Kenya Medical Research Institute; LMIC, low- and middle-income country; MEMS, micro electrical mechanical system; PPG, photoplethysmograp.

associated with weather variables—as we gather them from weather stations that we have setup as part of this study to cover the whole Nouna and Siaya HDSSs—in order to explore changes in behavior and other variables, such as activity, sleep, hr, during extreme weather events like heat stress exposure. Furthermore, wearable data could be linked to health outcomes and weather events. As a result, consumer-grade wearables may serve as a supporting technology for generating reliable measurements in low-resource contexts and investigating key links between weather occurrences and health outcomes. Thus, wearable devices may provide insights to better inform mitigation and adaptation interventions in these low-resource settings that are direly faced by climate change-induced changes, such as extreme weather events.

# Introduction

## Rationale and background

Health and demographic surveillance systems (HDSSs) have so far been used to understand population health outcomes in many resource-constrained settings. An HDSS is a geographically defined part of the population that is continuously monitored to promote collection and provision of timely data in vital events such as birth, death and (in- and out-) migration [1, 2]. The International Network for the Demographic Evaluation of Populations and Their Health (INDEPTH) network is the umbrella network for these HDSS and has grown to comprise over 3,800,000 people within 49 HDSS in 19 different countries on three continents [2]. Especially where reporting of critical events and health information systems are weak or non-existent, the HDSS offers a well-structured mechanism for collecting accurate, reliable population-based data. Each HDSS covers a dynamic population cohort that often changes based on entry (birth, in-migration) and exit (death, out-migration) events [2]. Through regular HDSS cohort monitoring with subsequent data collection rounds within the HDSS, longitudinal relational databases are built up of individuals and social units. As the epidemiological transition progresses in sub-Sahara Africa (SSA), life lived with disease, especially non-communicable disease, is increasing [3]. To understand the most pressing health needs of populations, it is key to also have reliable, accurate and consistent local burden of disease data [4]. Particularly, as the risk of climate change-induced extreme weather events increases, health effects are projected to increase dramatically in the next decades, especially in SSA [5–8]. The public health impact of climate change has been understudied and poorly conveyed as part of the larger effort to comprehend the ramifications of current and future changes in the global climate [9]. For example, a significant part of the population in SSA relies on rainfed subsistence agriculture and is thus vulnerable to extreme weather events [10], which is projected to be more deadly in SSA contexts than in other regions of the world [11]. Physiological and clinical impacts of heat stress have been described elsewhere in detail [12, 13]. However, reliable, accurate, consistent and objectively measured health outcomes continue to be limited due to a lack of appropriate measurement devices, manpower, and financial resources resulting in a dearth of longitudinal data. Particularly missing are high spatial geographical resolutions of weather data [14] (particularly wet bulb globe temperature, WBGT, an aggregated measure for heat stress exposure [15]) and individual data on activity and vital signs, like hr. Despite efforts such as the HDSS, it has to be noted, that reliable primary data on health and its environmental and social determinants are still limited and underutilized [1, 16, 17]. Resultantly secondary data from health services are potentially biased, incomplete and of low quality/validity [4].

Notably, self-reported research has generated many insights, including the HDSS, verbal autopsy amongst others, which proved to be an effective way of filling data gaps [1, 18].

Technological advances, such as non-invasive, consumer-grade wearable devices and their application in population-based studies [19–21], present an opportunity to gain insights on the individual level. Wearable devices track vital signs in the study participants' environment, the so called ecological momentary assessment [22], and allow insights into hr and general activity, oftentimes providing further insights like sleep activity, body temperature, global positioning system (GPS) and electrocardiography (ECG) data [22, 23]. In most cases, consumer-grade wearables provide a software landscape with application programming interfaces (APIs) that can be used right out of the box, without having to invest into software development. Furthermore, many consumer-grade wearables employ direct data transmission via mobile data networks to a central, local server, allowing data to be accessed instantly or in real time. The use of wearable devices for research is increasing, particularly as more accessible and affordable options become available [24], and rates of clinical approval of respective regulatory bodies rise [25, 26].

Studies have shown the potential of such consumer-grade wearable devices for population-health contexts [27]. For example, Culp and Tonelli et al. (2019) conducted a study on heat stress exposure using wearable devices. Their research focused on agricultural workers' physiological responses during agricultural work and linked them to working intensity and climate conditions (measured by the WBGT) [28]. The authors discovered considerably higher body temperatures, increased hr, and respiration rate in the uncomfortable intensity category. Lam et al. (2021) explored the association between short-term physiological and psychological thermal adaptation and outdoor thermal comfort in people who exercised in different temperature zones in China and found that addressing thermal discomfort early on can help prevent more serious heat-related disorders [29]. First studies employing wearable sensor in the field started approximately two decades ago in the military area [30–34]. However, most studies were conducted in high-income settings, like the US and Europe [19, 25, 35]. Consumer-grade wearable devices may provide a novel method to generate objective, highly-resolved individual data monitoring physiological parameters, such as hr, physical activity and sleep duration. This data may facilitate further research to consider aspects of climate change-induced extreme weather events on health outcomes in populations in low-resource contexts [36–38].

## Study goals and objectives

The overall objective of this study is to elucidate whether consumer-grade wearable devices are an acceptable and feasible means to generate data in low-resource contexts. Furthermore, we evaluate the validity of consumer-grade wearable devices in comparison to gold standard wearable devices. We will conduct studies in two HDSS sites in low-resource contexts: (i) in the Nouna HDSS managed by the Centre de Recherche en Santé (CRSN) and (ii) the Siaya HDSS managed by the Kenya Medical Research Institute (KEMRI). The study is conducted in the two African countries of Burkina Faso and Kenya, as both countries are located in SSA and have one of the highest burdens of climate-sensitive diseases, but differ in their climatic, socio-economic and disease profile [2, 39–41].

The specific research questions are:

1. are consumer-grade wearables a feasible and acceptable means to generate vital sign measurements in a rural population in low-resource contexts of Burkina Faso and Kenya?

2. is it feasible (a) to generate insights on individual behavior with a recall activity journal and (b) to identify pre-defined activities based on this data?

3. when compared to gold-standard sensors, can consumer-grade wearables produce valid measurements in rural Burkina Faso and Kenya?

We will investigate the feasibility and acceptability with a Likert-scaled questionnaire that covers also demographic items: 13 items addressing feasibility with a strong focus on hardware reliability (i.e., are the devices functioning well in the sub-Saharan climatic conditions), and 29 items addressing acceptance (see S1 File for the questionnaire). If the wearable devices fail or we have data outages, we shall keep track of them in order to gain a quantitative overview of the types of failures and sources of data disruptions.

The validity of wearable devices is compared to gold sensor devices, further described in the section *Validation study*. The wearable devices are described in the section *Study technology*.

As part of this study, we have installed ten fully automated weather stations with the same configuration in the Nouna HDSS (5x) and in the Siaya HDSS (5x) that spatially cover the whole HDSS area. As a follow-up study to this feasibility, acceptability and validation study, it is planned to generate wearable measurements in a one-year study to account for seasonality and investigate whether weather (mainly based on WBGT) has an effect on individual behavior and health (climate-sensitive diseases, as available within the Nouna and Siaya HDSS). The local weather stations that have been setup as part of this research unit cover the continuous measurements of precipitation, wind speed, wind direction, radiation, temperature, and humidity, as well as WBGT. Seasonality has no bearing on the feasibility, acceptability, or validation of the wearable measurement in this study, hence it is not taken into consideration.

This study protocol is reported according to the World Health Organization recommended format for a 'research protocol' [42].

## Materials and methods

### Study design

**Feasibility and acceptability study.**   The study will take place in two countries, respectively in the Nouna HDSS (n = 150) and in the Siaya HDSS (n = 150) with a total of n = 300 study participants using consumer-grade wearable devices (see section *Study participants and sample size* for sample size calculation, inclusion and exclusion criteria). Study participants will be interviewed three times during the study cycle (see S1 for questionnaire, see Fig 1 for details to study cycle), once each week.

A trained nurse or field worker will conduct face-to-face interviews with study participants using tablet-based questionnaires, which comprises four parts:

1. personal information of study participants (6 items): study ID, village name, gender, date of birth, weight, height

2. distribution and management by the field workers of wearable devices (13 items per wearable device): device returned or handed out (date), details of device (ID, wearable type), disinfection of wearable device, system registration of study participant, reporting of device damages, reported acceptance of study participant

3. acceptance (29 Likert-scaled items): perceived ease of use, self-efficacy, perceived enjoyment, anxiety, experience (variables based on Technology Acceptance Model [43])

4. activity journal (structured activity list, reflecting 21 most common local activities): study participants report retrospectively to the field worker or study nurse their activities the day prior to the questionnaire, respectively to the time frames: after getting up in the morning, during the morning, midday, afternoon, evening, night

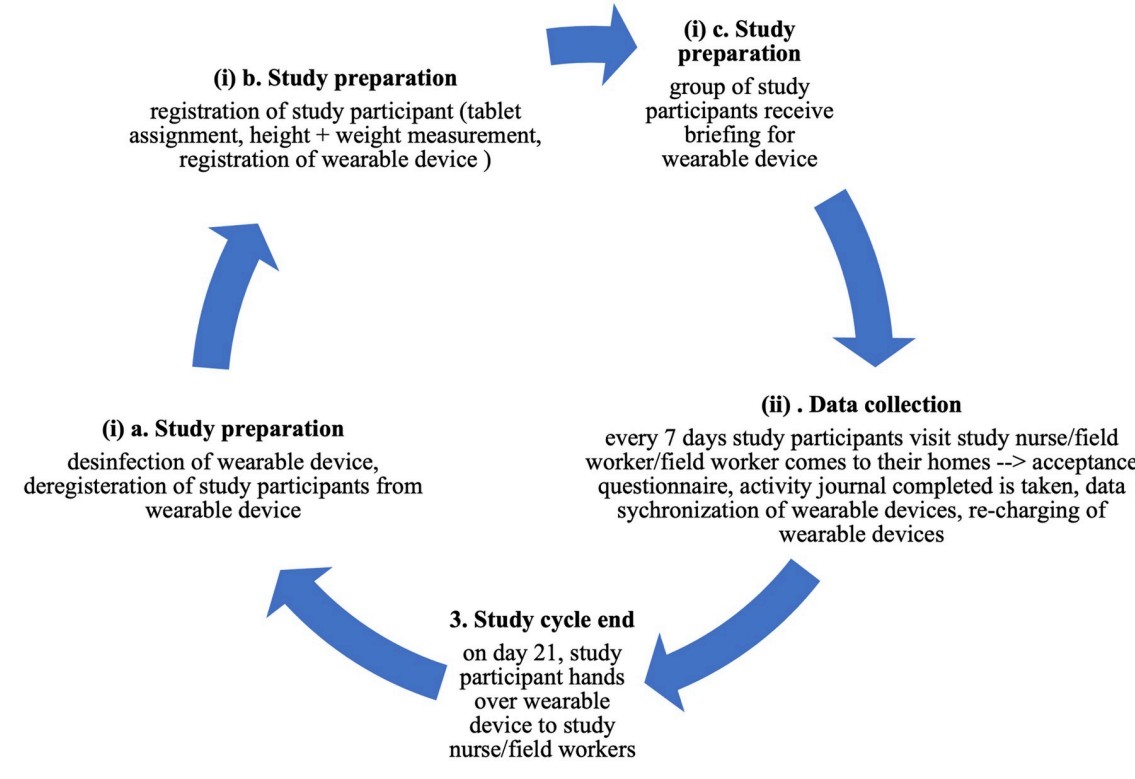

**Fig 1. Overview cycle of wearable device feasibility study.** The study cycle depicts the individual steps that are conducted, whereby a total of three study cycle will be conducted that comprise each n = 50 study participants.

The wearable outcomes of the study comprise: hr, sleep duration, activity (steps per day) and body shell temperature. Further quantitative components include: (i) the wearable device data, (ii) comparison data between the wearable devices and gold standard wearable devices (see section *Study Technology* for details).

Data collection will take place over a nine-week period which comprises three study cycles, where each cycle lasts three weeks comprising n = 50 study participants per study site (see section *Study participants and sample size* for sample size calculation). We obtained only a limited number of wearable devices (n = 50 Withings Pulse HR; n = 25 Tucky thermometer), as we were uncertain of the study outcome and did not want to exhaust financial resources. We will request study participants to give their consent before study participation and will exclude them otherwise. The details of the study cycles will be as following (see Fig 1 for details):

1. At first, the study nurse (study arm 1) or field worker (study arm 2 and 3) hands out the disinfected wearable devices to the study participants, based on their allocation to the respective study arm, study participants will receive (a) only the Withings Pulse HR, or (b) the Withings Pulse HR and Tucky thermometer. To decrease the likelihood to transmit infectious diseases like the Coronavirus disease (Covid-19), each study nurse and community field worker is equipped with a hygiene package comprising face masks, disinfectant spray, clean wipes and hand disinfectant. The wearable devices are first rinsed with water, then sprayed with disinfectant, and wiped clean. A protocol that details this process will be handed out to each study nurse and field worker. The wearable devices will be registered to the respective study participant.

2. Study participants will be trained on critical aspects about wearing and using the wearable device, such as battery charging, as well as receive a comprehensive introduction of the study's procedures.

3. Every 7 days, we will collect data from study participants. Based on their study arm allocation participants will be visited by the study nurse (study arm 1), by the field worker (study arm 2), or will be visited by the field worker at home (study arm 3). In study arm 1 and study arm 2, we will administer an acceptance questionnaire, collect hr, activity and sleep quality via the Withings Pulse HR and body shell temperature with the Tucky wearable device, whereby wearable data is synchronized with a tablet, and recharge the wearable device. Furthermore, the study nurse or the field worker (respective to their study arm) will fill out a structured activity journal in which the study participant will be asked about their activities (structured list) from the previous day (from the time they got up from bed to going back to sleep at night). In study arm 3, we will train participants to synchronize both the Withings Pulse HR and Tucky wearable device themselves with a smartphone, and recharge the battery of the wearable devices using a foldable, portable solar panel with a USB-port which are provided to study participants as part of the study.

4. On day 21, the study participant will return the wearable device to the study nurse or field worker, who will then disinfect and deregister the wearable device. The study nurse or field worker will then assign the wearable device to the next research cycle's study participant.

The questionnaire will be hosted on our local research server using SurveySolutions software, a data collection and management tool (https://mysurvey.solutions/en/).

**Validation study.** We will compare the consumer-grade to the gold-standard wearables during the last study cycle (week 6 to week 9). The validation study is divided into two runs of two weeks each. Per location, we will select a sub-sample of study participants of n = 20 (n = 10 females, n = 10 males), spanning over all study arms. During each run, study participants will wear both consumer-grade and validated gold standard wearables (see section *Study Technology* for details), which are detailed in full below:

1. For two weeks:

   - 24/7: (i) wearable Withings Pulse HR (consumer-grade), (ii) high-resolution wrist accelerometer (GENEActiv Original watch, ActiveInsights, UK, resolution up to 100Hz, which is a raw data actigraphy data logger that records a digitally integrated measure of gross motor activity as well as sleep schedule variability, sleep quantity, and sleep quality statistics and daytime), (iii) actigraphy-based data logger

   - during nights only (as device may not remain on the body during day-time activities): wearable thermometer Tucky (consumer-grade)

2. For 24 hours (starting after day 12 of GENEActiv wearing)

   - a wearable one-lead electrocardiography (ECG) device (Bittium Faros, Finland, sample rate 250 Hz)

   - a validated heat-flux sensor (Tcore, Dräger, Germany), connected to a miniaturized data logger (HFUM, KORA Industrie Elektronik, Germany) [44–48], both integrated in a custom-made headband, to be worn comfortably.

Specifically, data collected with Bittium faros ECG will serve as a reference for the hr measurements done with the Withings HR data. We will compare the raw data collected with the wrist accelerometer GENEActiv with the Withings Pulse HR data (hear rate, activity data), and

furthermore we will compare data collected with the Tcore sensor (core body temperature) with the Tucky thermometer data (shell body temperature).

## Study participants and sample size

**Feasibility and acceptability study.**   We will sample a total of n = 300 study participants for the feasibility and acceptability study (Nouna HDSS n = 150; Siaya HDSS n = 150). The sampling will be randomly drawn from the respective HDSS population, whereby the study population is stratified according to age (three groups: 6-16yrs, 17-45yrs, >45yrs) and gender (n = 150 females, n = 150 males).

Study participants are eligible for the study if individuals: (i) are residents within the Nouna and Siaya HDSS, (ii) are 6 years of age or above (as wristband size of the Withings Pulse HR is not deemed adequate for children younger than 6 years), (iii) are willing to use a Bluetooth or mobile data-enabled mobile device for research. After identification, the local study teams in the Nouna and Siaya HDSS will approach study participants face-to-face. In case an identified study participant will not want to participate in the study, we will draw a sample of n = 170 per HDSS, to have alternate study participants at hand.

Sample size calculation (survey/questionnaire) to comprise n = 150 study participants/ HDSS was calculated based on a population size of n = 100.000 (total HDSS population excluding children under the age of 6 years), a confidence level of 95% and a margin of error of 8%.

For generating insights with the wearables devices data, we deemed n = 150 as sufficient, as reported in the literature [49].

For data collection, we will recruit a total of five interviewers, including two health workers and three field workers. All the interviewers will have strong expertise in electronic data collection and will be overseen by a statistician and a computer scientist with extensive experience in managing the HDSS.

Validation study. For the validation study, considering as a first main outcome HR, a sample size of n = 20 participants is deemed sufficient, as reported in the literature [50–52]. Here, we selected a total of n = 20 men and n = 20 women, considering both the HDSS in Nouna and Siaya (n = 10 females, n = 10 males per HDSS).

People will be eligible for the validation study if: 18–45 years of age, living and working in the area surrounding a health care facility (in walking distance), absence of any medical condition that would interfere with the data collection, such as for example cardiovascular or metabolic diseases (hypertension, diabetes mellitus, overweight or obesity), renal diseases as well as absence of acute malaria infection or other infectious diseases (such as HIV, tuberculosis, streptococcus, human papillomavirus, viral hepatitis and parasitic diseases).

Field workers and local study managers in Burkina Faso and Kenya will inform study participants about research details and proceedings. Only if study participants agree to participate in the study, they sign an informed consent form. There will be no personal relations between the field workers and study managers. However, some may be familiar with some study participants from prior studies. The respective HDSS will recruit field staff and interviewers. The study participants are reimbursed for their time commitment in the study and the type and scope are subject to local HDSS guidelines.

## Study technology

**Consumer-grade wearable devices.**   For this study, we will use the data collection platforms provided through the wearable devices' manufacturers. A number of variables will be collected from each study participant throughout the study period, comprising the following:

Table 1. Overview of consumer-grade wearable devices.

| | Withings Pulse HR | Tucky Thermometer |
|---|---|---|
| measures | • **steps (distance and calories):**<br> • measured continuously, steps identified based on amplitude and periodic pattern<br> • technology: accelerometer<br>• **heart rate:**<br> • routinely measured every 10 mins<br> • measurement frequency every 1 sec (continuous heart rate mode) only in workout session or after 2 mins of running<br> • technology: photoplethysmography<br>• **sleep**<br> • tracked based on sleep score, 4 inputs: duration (total time spent sleeping), regularity (consistency between bed- and rise-times), depth (part of night spent in deep sleep), interruptions (time spent awake) | • body temperature (shell temperature)<br> • measured continuously with contact sensor<br> • every 60 sec one measurement<br> • fever alert (default: 38,4˚C, adjustable)<br>• position tracking (not used in this study)<br> • real time sleeping / lying position<br> • 3 categories: on the back, on the stomach or on the side<br>• technology: accelerometry |
| wear location | wrist | under right armpit |
| wear frequency | during the whole study cycle | during night |
| data synchronization | 5 days of local storage of data between synchronizations (within 10m of tablet) | requires regular synchronization (within 10m of tablet) |
| charging | up to 21 days | 5–7 days |
| sensor | • high precision micro electrical mechanical system (MEMS) 3-axis accelerometer<br>• photoplethysmography sensor | • Thermistor contact thermometer sensor measuring the average temperature between the sensor and the skin surface [53]<br>• 3- and 1-axis accelerometer |
| connectivity | • Bluetooth low energy | • Bluetooth low energy |

hr, steps, sleep, weight, height, and shell body temperature (see Table 1 for details on consumer-grade wearables).

**Gold-standard wearable devices.** According to the three main outcomes: hr, physical activity and core body temperature (CBT), we identified gold-standard wearable devices who will provide a reference measurement which we will compare with the consumer-grade wearable devices. For precisely measuring hr, we will continuously collect 24 hour one-lead ECG (sample rate 250 Hz), accounting for ectopic beats and arrhythmic events. We will continuously monitor activity with a validated wrist-worn tri-axial accelerometer (resolution up to 100Hz) device (GENEActiv Original, ActiveInsights, UK). The accelerometers are small, rugged, waterproof, actigraphy-based data loggers that record a digitally integrated measure of gross motor activity, as well as sleep schedule variability, sleep quantity, and sleep quality statistics and daytime [54–58]. In addition, the systems are equipped with luminous flux recording, to distinguish between time spent outdoors and indoors, and near body temperature sensor, to define precisely wearing time.

CBT will be continuously recorded for 24 hours by placing a non-invasive technology on the forehead. The double heat-flux sensor technology has been successfully tested during hypothermia, bedrest, and exercise on Earth and in space [44–48]. Based on this technology, for this study, we implement the Tcore™ sensor (Dräger, Lübeck Germany), a recently developed a new disposable, soft sensor which perfectly adapts to the forehead shape, in order to improve comfort and wearability. Briefly, the Tcore™ employs a non-invasive technology where a unique dual-sensor heat flux system accurately and continuously calculates CBT following a short ramp-up time. This single-use sensor is connected via cable to a miniaturized data logger (KORA Industrie Elektronik, Hambühren, Germany), that can be comfortably

**Table 2. Overview of gold standard wearable devices.**

| Gold standard devices | ECG monitor | Wrist worn actigraphy | CBT sensor, data logger headband |
|---|---|---|---|
| measures | • One-lead continuous electrocardiography<br>• tri-axial accelerometer | • Physical activity, sleep efficiency, energy expenditure based on tri-axial accelerometers raw data<br>• light exposure<br>• near body temperature | • core body temperature continuously |
| size | 48 x 29 x 12 mm, weight ca 16 g | 43 x 40 x 13 mm, weight ca 16 g (without strap) | Sensor: 60 x 50 x 4 mm Data logger: 48 x 30 x 5 mm, weight ca 15 g |
| wear location | Thorax: 3 electrodes | Wrist (non-dominant) | Forehead (headband) |
| wear frequency | 24-h | 2 weeks continuously | 24-h |
| data synchronization | 24-h data internal storage, synchronization time: 5 min | 2-week data internal storage, synchronization time:5 min | 24-h data internal storage, synchronization time: 5 min |
| charging | Up to 7 days | Up to 45 days | Sensor: 24-h replacement Data logger: up to 60 days |
| sensor | ECG sensor (Faros 180, Bittium, Finland, up to 1000 Hz sample rate) | high precision MEMS, resolution 3.9mg, up to 100 Hz sample rate (GENEActiv Original, ActiveInsights, UK) | Unique dual-sensor heat-flux technology (Tcore™, Dräger, Germany) |
| connectivity | USB cable | Cradle with USB cable | USB cable (data logger) |

worn with custom-made headband, which integrates both sensor and data logger (see Table 2 for overview of gold standard wearable devices).

## Setting

The Nouna HDSS, Burkina Faso [59], and the Siaya HDSS, Kenya [40], provide access to comprehensive, retrospective health and population data comprising of nearly 20 years of data and more than 260,000 people under surveillance. The surveillance area of the Nouna HDSS is characterized by a tropical climate with one rainy season usually lasting from May to September (mean annual rainfall of 800mm) and overall high temperatures throughout the year. Malnutrition and malaria are both common in the Nouna HDSS. In the surveillance area of the Siaya HDSS, the climate is tropical with two annual rain seasons, with the heaviest long rains usually occurring from March through May and short rains falling between October and December (mean annual rainfall of 1200mm).

We will conduct the study in two HDSS, as the two study sites are quite different in their climatic, socioeconomic and disease profile. Due to this unbalanced distribution of diseases, as well as different available levels of adaptation and mitigation measures, population impacts of climate change on health are likely to vary considerably between and within these countries.

## Analysis methods

**Feasibility of using wearable devices for populations in low-resource contexts.** We will use a deductive thematic analysis approach to analyze the feasibility questionnaire, grouping data into pre-defined categories and expanding categories if additional themes arise from the data, mainly from the free text fields of the questionnaire. To investigate disruptions in the wearable data generation, we will link the questionnaire data with the collected records of wearable device disfunctions and data interruptions.

**Feasibility to generate insights on individual behavior.** As a first explorative approach, we will extract activity patterns as collected with the wearable devices and correlate them to self-reported activity, that we will collect with the retrospective recall diary (for each study participant we will have three full days of reported activates). This is a first exploration to

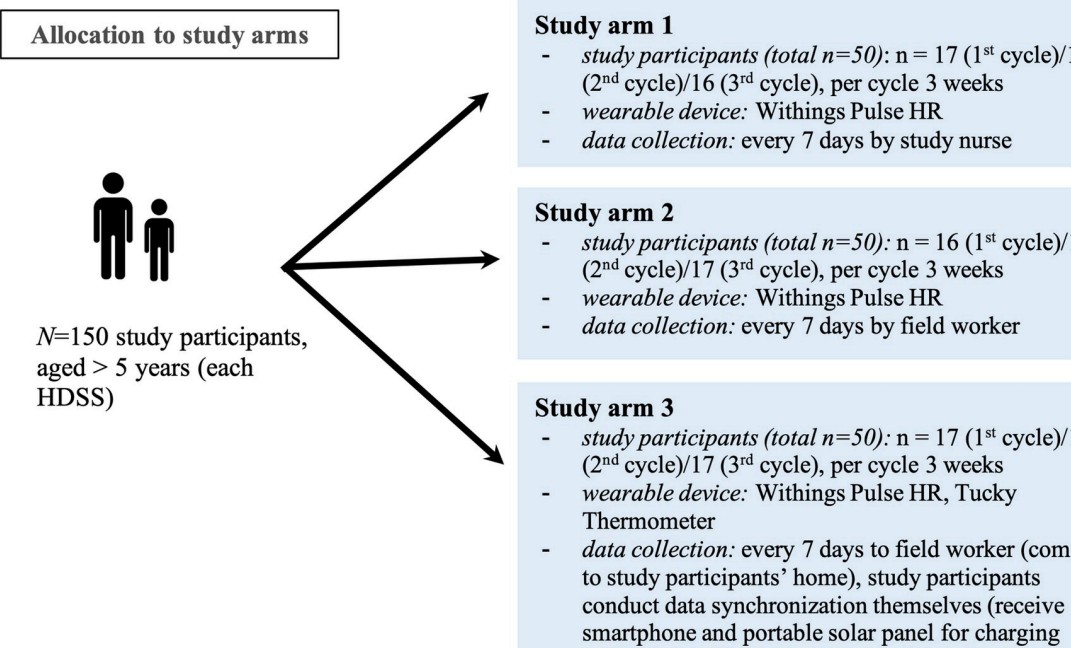

**Fig 2. Three study arms in detail.** Allocation of study participants to different study arms.

investigate the feasibility of classifying common local activities like harvesting, fetching water and so on, using time-domain plots of wearable device data, as reported in other research [60].

Furthermore, we will investigate correlations of weather events on hr, activity, sleep, and body shell temperature. We will run a multiple linear regression with hr, activity, sleep, and body shell temperature as dependent variables and WBGT, precipitation, air temperature, relative humidity, solar radiation, and wind speed as independent variables. The weather data will be taken from the ten local weather stations that spatially cover both HDSS areas in the Nouna HDSS and Siaya HDSS.

**Acceptance.** The analysis of the acceptance questionnaire (based on 5-point Likert-scale, questions will be asked for level of agreement from *strongly agree* to *strongly disagree*) is conducted with a parametric analysis of variance (ANOVA) to compare means of the three study arms (see Fig 2 for details). The questionnaire items are grouped according to the employed Technology Acceptance Model constructs [43].

Baseline data will be reported for all three arms and summarized as mean (standard deviation, SD) or median (first quartile, third quartile) for continuous variables and as count and number (percentage) for categorical variables. Furthermore, we will calculate hourly, daily, and weekly means (SD) of hr, activity and sleep measures, as well as body shell temperature. Demographic data is descriptively analyzed in R with graphs for visualization (version 4.0.3; The R Foundation for Statistical Computing, Vienna, Austria). Collected data on the Withings' Health Mate platform (variables: hr, steps, sleep of the Withings Pulse HR; weight, height), and on the Tucky E-Takescare platform (shell body temperature) will be exported, and then cleaned and analyzed with the statistical software R. The data (hr, activity, sleep, if available: body shell temperature) between the Nouna and the Siaya HDSS will be compared via ANOVA.

**Validation.** As for the validation study, we will assess the agreement between different methods—the difference between outcome values from consumer-grade devices and the respective values measured with gold-standard devices–following the Bland-Altman-plot [61] and the Lin Concordance correlation coefficient [62].

## Ethics and consent

The study protocol was approved by the Kenya Medical Research Institute (approval date: 11th March, 2020; KEMRI/SERU/CGHR/327/3962), the Comité d'ethique pour la recherche en santé in Burkina Faso (approval date: 13th March, 2020; 2020-3-041), the ethical committee of the University Hospital Heidelberg (approval date: 6th May 2019; S-294/2019), Germany and the ethical commission of Charité, Berlin, Germany (approval date: 11th March 2019; EA1/060/19).

## Expected outcomes of the study and discussion

Our study will evaluate the feasibility, acceptability and validity of consumer-grade wearables in the low-resource context of Burkina Faso and Kenya. We will generate critical insights on whether consumer-grade wearables that measure hr, activity (steps/day), sleep duration and body shell temperature can be used to generate valid and reliable data on an individual level in low-resource context. Wearable devices can be deemed unreliable, in particular when human activity impedes measurement [63]. A study has found that substantial differences exist between various devices and various activities, at times showing significantly high average error as compared to measured resting periods [64]. Wearables accuracy at rest and during physical activity differ, and we will generate insights by comparing consumer and research-grade wearables in this context. To understand the acceptability, feasibility and validity of wearable devices is vital and their ability to produce reliable data. In particular, objectively measured health outcomes continue to be limited due to a lack of appropriate measurement devices, manpower, and financial resources resulting in a scarcity of longitudinal data. We will also investigate whether wearables are reliable devices in these almost extreme environments of SSA, as well as whether they can provide insights into individual behavior and patterns in terms of activity, hr, body shell temperature and sleep, as these are the primary variables recorded by the wearable devices that we will employ in this study.

Furthermore, we will share our shortcomings and benefits for using wearable devices for research in low-resource contexts. We will detail participants' acceptance of wearable devices, and highlight the strategies for extracting data and managing wearable devices to ensure sustained use of devices. Currently, there is a paucity of research on the usability of wearable devices in low-resource contexts.

## Conclusion

Our study will contribute to the general scientific body of knowledge for using consumer-grade wearable devices to measure individual vital signs in a low-resource research context. Particularly, our anticipated objective is to employ these consumer-grade wearables devices for climate change and health research, understanding, for example, the impacts of weather events such as heat and work productivity. Low-resource countries such as Burkina Faso and Kenya are predicted to be most vulnerable to climate change, as the burden of disease of climate-sensitive health outcomes is projected to increase considerably within the next decades [7]. Therefore, consumer-grade wearables may constitute a novel way to explore critical relationships between weather events and health outcomes. Furthermore, our research may be able to bridge the gap in the current lack of evidence since oftentimes research infrastructures

supporting health research in SSA, especially in health and demographic surveillance systems (HDSSs), are not equipped to investigate many of the most pressing climate change and health research needs [65].

## Supporting information

**S1 File. Feasibility and acceptability questionnaires, and retrospective activity diary.** (PDF)

**S2 File. Informed consent forms.** (PDF)

## Author Contributions

**Conceptualization:** Sandra Barteit, Valentin Boudo, David Obor, Rainer Sauerborn, Martina A. Maggioni, Till Bärnighausen.

**Formal analysis:** Sophie Huhn, Martina A. Maggioni.

**Funding acquisition:** Sandra Barteit, Ali Sié, Stephen Munga, Aditi Bunker, Rainer Sauerborn, Till Bärnighausen.

**Investigation:** Sandra Barteit, Valentin Boudo, David Obor, Martina A. Maggioni.

**Methodology:** Sandra Barteit, Aditi Bunker, Rainer Sauerborn, Hanns-Christian Gunga, Martina A. Maggioni, Till Bärnighausen.

**Project administration:** Sandra Barteit, Valentin Boudo, Aristide Ouedraogo, Pascal Zabré, Ali Sié, Stephen Munga, David Obor, Daniel Kwaro.

**Resources:** Sandra Barteit, David Obor, Daniel Kwaro, Till Bärnighausen.

**Supervision:** Sandra Barteit, Lucienne Ouremi, Ali Sié, Stephen Munga, David Obor, Rainer Sauerborn, Till Bärnighausen.

**Validation:** Sandra Barteit, Martina A. Maggioni.

**Writing – original draft:** Sandra Barteit, Sophie Huhn, Martina A. Maggioni.

**Writing – review & editing:** Sandra Barteit, Valentin Boudo, Aristide Ouedraogo, Pascal Zabré, Lucienne Ouremi, Stephen Munga, David Obor, Daniel Kwaro, Sophie Huhn, Aditi Bunker, Rainer Sauerborn, Hanns-Christian Gunga, Martina A. Maggioni, Till Bärnighausen.

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
