## [Decision Letter · Decision Letter 0]

14 Jul 2021

PONE-D-21-17308

Feasibility, acceptability and validation of wearable devices for climate change and health research in the low-resource contexts of Burkina Faso and Kenya: Study protocol

PLOS ONE

Dear Dr Barteit

I apologize for the delay, we have received the response of the second reviewer.

Thank you for submitting your manuscript to PLOS ONE. After careful consideration, we feel that it has merit but does not fully meet PLOS ONE’s publication criteria as it currently stands. Therefore, we invite you to submit a revised version of the manuscript that addresses the points raised during the review process.

In accordance to the other colleagues, your manuscript need of minor revisions and should be modified as suggested.

best regards

manuela cabiati

We look forward to receiving your revised manuscript.

Kind regards,

Manuela Cabiati, Ph.D.

Academic Editor

PLOS ONE

Journal Requirements:

2. Please include the additional information regarding ethics approval in the 4 June 2021 email in the Response to Reviewers.

5. Please ensure that you refer to Figure 1 in your text as, if accepted, production will need this reference to link the reader to the figure.

Reviewers' comments:

Reviewer's Responses to Questions

**Comments to the Author**

1. Does the manuscript provide a valid rationale for the proposed study, with clearly identified and justified research questions?

Reviewer #1: Yes

Reviewer #2: Yes

2. Is the protocol technically sound and planned in a manner that will lead to a meaningful outcome and allow testing the stated hypotheses?

Reviewer #1: Yes

Reviewer #2: Yes

3. Is the methodology feasible and described in sufficient detail to allow the work to be replicable?

Reviewer #1: Yes

Reviewer #2: Yes

4. Have the authors described where all data underlying the findings will be made available when the study is complete?

Reviewer #1: Yes

Reviewer #2: Yes

5. Is the manuscript presented in an intelligible fashion and written in standard English?

Reviewer #1: Yes

Reviewer #2: Yes

6. Review Comments to the Author

You may also provide optional suggestions and comments to authors that they might find helpful in planning their study.

Reviewer #1: This is an interesting manuscript that comprises a study protocol for the purpose of conducting a climate change and health research in low-resource areas using wearable devices. Overall, the protocol looks fine and the expected results would be valuable.

I don’t have major objections to the manuscript. I do have the following small issues for the authors to consider:

1. I suggest the authors to consider is it necessary to monitor all the devices to see if some are not working properly (e.g., if some of them are under direct sunshine and the battery got burned) and what to do if malfunctions do occur.

2. I’m not sure if (regular) incentives are necessary to provide to the participants to finish the study.

3. I’m not sure if the data would be enough for the authors to use deep learning models for analysis.

Reviewer #2: This manuscript presents the study protocol to investigate the acceptability, feasibility and validity of some wearable devices to study a sub-Saharan Africa population subjected to heat stress. The presentation of the protocol is well described and the number of subjects who will be enrolled seems to be compatible with significant statistical power for the subsequent analysis.

The issue is certainly important considering the problem of climate change that we are experiencing in this era.

Perhaps it would be appropriate to specify the method of sterilization of the instrumentation supplied in the study also in light of the problems of Covid 19.

A likely critical point within the various physiological parameters that will be monitored is the measurement of blood pressure, which is only foreseen at the time of the meeting with the nurse. This parameter will certainly not give information on the relative trend of blood pressure so it could be useful if a measurement within 24 hours of the subjects enrolled could be useful to be related to the environmental climatic characteristics.

Considering also a possible purpose of the study which could be to evaluate the cardiovascular response to heat stress, it is also asked to specify how the environmental climatic parameters will be detected during the enrollment period of the subjects.

7. PLOS authors have the option to publish the peer review history of their article (what does this mean?). If published, this will include your full peer review and any attached files.

Reviewer #1: No

Reviewer #2: **Yes: **Lorenza Pratali

---

## [Author Response · Author response to Decision Letter 0]

22 Jul 2021

resp. 1. We have revised the submitted manuscript according to the PLOS ONE’s style requirements and I hope that we have adhered to all style requirements now in our revised manuscript. 

resp. 2. I have uploaded all research ethics committee approval letters (Heidelberg University Hospital, SERU Kenya, CRSN Burkina Faso, Charité Berlin, uploaded in the Editorial Manager on May 28th, 2021). As per our study protocol (CP1) we collaborate with sub-project P4 as part of this research unit, as we compare the consumer-grade wearable devices that are part of sub-project CP1 with the gold standard sensors of sub-project P4 (Charité). The Charité ethical approval is not a necessary element for the study protocol of CP1, as we have ethical approval from Heidelberg University, the CRSN in Burkina Faso and the SERU in Kenya. 

With regards to the SERU ethical approval: The SERU in Kenya has approved all sub-projects that are part of this research unit in one approval of the complete DFG research unit "Climate Change and Health in Sub-Saharan Africa". Prof. Dr. Dr. Rainer Sauerborn is the spokesperson of this research unit and has such is addressed in the SERU approval letter. I hope this can clarify on the respective ethical approvals.

resp. 3. We have revised the funding information in the manuscript accordingly. 

resp. 4. We have moved the ethics statement to the Methods section.

resp. 5. We revised the manuscript so all figures are referred to in the text. Furthermore, we have uploaded our two figures as included in this revised manuscript to PACE.

resp. 6. We have added the following references to the revised manuscript: 

- we have added the following reference as we have added further examples of research that has already used wearable devices data correlated with weather data

28. Culp K, Tonelli S. Heat-Related Illness in Midwestern Hispanic Farmworkers: A Descriptive Analysis of Hydration Status and Reported Symptoms. Workplace Health Saf. 2019;67: 168–178. doi:10.1177/2165079918813380

- we have added the following reference as we have added further examples of research that has already used wearable devices data correlated with weather data

29. Lam CKC, Hang J, Zhang D, Wang Q, Ren M, Huang C. Effects of short-term physiological and psychological adaptation on summer thermal comfort of outdoor exercising people in China. Build Environ. 2021;198: 107877. doi:10.1016/j.buildenv.2021.107877

- we have added the following reference to complete the statement in the sentence in the manuscript (p.8): The study is conducted in the two African countries of Burkina Faso and Kenya, as both countries are located in SSA and have one of the highest burdens of climate-sensitive diseases, but differ in their climatic, socioeconomic and disease profile [2,39–41].

41. Sewe M, Rocklöv J, Williamson J, Hamel M, Nyaguara A, Odhiambo F, et al. The association of weather variability and under five malaria mortality in KEMRI/CDC HDSS in Western Kenya 2003 to 2008: a time series analysis. Int J Environ Res Public Health. 2015;12: 1983–1997. 

- we have added further specifications in the Analysis section of the manuscript (p.22) and have accordingly added this reference to refer to the specified research: “This is a first exploration to investigate the feasibility of classifying common local activities like harvesting, fetching water and so on, using time-domain plots of wearable device data, as reported in other research [60].”

60. Liu Y, Nie L, Liu L, Rosenblum DS. From action to activity: Sensor-based activity recognition. Neurocomputing. 2016;181: 108–115. doi:10.1016/j.neucom.2015.08.096

We would like to thank the reviewers for their comments and helpful suggestions to our manuscript. Accordingly, we have made amendments to the manuscript and describe our amendments with respect to the reviewers’ comments in the following. 

Reviewer #1: This is an interesting manuscript that comprises a study protocol for the purpose of conducting a climate change and health research in low-resource areas using wearable devices. Overall, the protocol looks fine and the expected results would be valuable.

I don’t have major objections to the manuscript. I do have the following small issues for the authors to consider:

1. I suggest the authors to consider is it necessary to monitor all the devices to see if some are not working properly (e.g., if some of them are under direct sunshine and the battery got burned) and what to do if malfunctions do occur.

- Many thanks for this comment. We do take up hardware reliability as part of this study and have added the following section to the manuscript, which we hope clarifies this point a bit further, p.8: “We will investigate the feasibility and acceptability with a Likert-scaled questionnaire that covers also demographic items: 13 items addressing feasibility with a strong focus on hardware reliability (i.e., are the devices functioning well in the sub-Saharan climatic conditions), and 29 items addressing acceptance (see S1 File for the questionnaire). If the wearable devices fail or we have data outages, we shall keep track of them in order to gain a quantitative overview of the types of failures and sources of data disruptions.”

2. I’m not sure if (regular) incentives are necessary to provide to the participants to finish the study.

- Indeed, study participants receive an adequate reimbursement (according to local standards). We have added on p.16 the following: “The study participants are reimbursed for their time commitment in the study and the type and scope are subject to local HDSS guidelines.”

3. I’m not sure if the data would be enough for the authors to use deep learning models for analysis.

- Again, many thanks for this comment. Indeed, this analysis is targeted for the subsequent one-year study that is planned to follow this study. We did not specify this well in the manuscript and after consideration have removed this paragraph, as it seems to be overburdening the current study protocol, which focus on feasibility, acceptability and validity.

Reviewer #2: This manuscript presents the study protocol to investigate the acceptability, feasibility and validity of some wearable devices to study a sub-Saharan Africa population subjected to heat stress. The presentation of the protocol is well described and the number of subjects who will be enrolled seems to be compatible with significant statistical power for the subsequent analysis.

The issue is certainly important considering the problem of climate change that we are experiencing in this era.

Perhaps it would be appropriate to specify the method of sterilization of the instrumentation supplied in the study also in light of the problems of Covid 19.

- Many thanks for this comment, which covers an important aspect. We have specified the sterilization in light of COVID-19 in the manuscript, p.10/11: “To decrease the likelihood to transmit infectious diseases like the Coronavirus disease (Covid-19), each study nurse and community field worker is equipped with a hygiene package comprising face masks, disinfectant spray, clean wipes and hand disinfectant. The wearable devices are first rinsed with water, then sprayed with disinfectant, and wiped clean. A protocol that details this process is handed out to each study nurse and field worker.”

A likely critical point within the various physiological parameters that will be monitored is the measurement of blood pressure, which is only foreseen at the time of the meeting with the nurse. This parameter will certainly not give information on the relative trend of blood pressure so it could be useful if a measurement within 24 hours of the subjects enrolled could be useful to be related to the environmental climatic characteristics.

- Yes, we agree that a 24hour measurement would be of great help to correlate this with weather characteristics. However, after consideration, we have removed the blood pressure measurement for this study, as we primarily want to focus on the feasibility, acceptability and reliability of the wearable devices and adding blood pressure at this point in time may only add further complications instead of insights. We will consider your remark for our one-year study that is to follow this study and we will aim for a 24-hour measurement to get higher granularity of individuals response to local weather events with regards to a blood pressure measurement. 

Considering also a possible purpose of the study which could be to evaluate the cardiovascular response to heat stress, it is also asked to specify how the environmental climatic parameters will be detected during the enrollment period of the subjects.

- We have detailed specifics on the ten local weather stations that we have setup as part of this research unit in Kenya and Burkina Faso, p.8/9: “As part of this study, we have installed ten fully automated weather stations with the same configuration in the Nouna HDSS (5x) and in the Siaya HDSS (5x) that spatially cover the whole HDSS area. As a follow-up study to this feasibility, acceptability and validation study, it is planned to generate wearable measurements in a one-year study to account for seasonality and investigate whether weather (WBGT) has an effect on individual behavior and health (climate-sensitive disease, as available within the Nouna and Siaya HDSS). The local weather stations that have been setup as part of this research unit cover the continuous measurements of precipitation, wind speed, wind direction, radiation, temperature, and humidity, as well as WBGT. Seasonality has no bearing on the feasibility, acceptability, or validation of the wearable measurement in this study, hence it is not taken into consideration.”

---

## [Editor Report · Decision Letter 1]

25 Aug 2021

Feasibility, acceptability and validation of wearable devices for climate change and health research in the low-resource contexts of Burkina Faso and Kenya: Study protocol

PONE-D-21-17308R1

Dear Dr. Barteit,

We’re pleased to inform you that your manuscript has been judged scientifically suitable for publication and will be formally accepted for publication once it meets all outstanding technical requirements.

Kind regards,

Manuela Cabiati, Ph.D.

Academic Editor

PLOS ONE
---

## [Editor Report · Acceptance letter]

3 Sep 2021

PONE-D-21-17308R1 

Feasibility, acceptability and validation of wearable devices for climate change and health research in the low-resource contexts of Burkina Faso and Kenya: Study protocol 

Dear Dr. Barteit:

I'm pleased to inform you that your manuscript has been deemed suitable for publication in PLOS ONE. Congratulations! Your manuscript is now with our production department. 

Kind regards, 

on behalf of

Dr. Manuela Cabiati 

Academic Editor

PLOS ONE